# Is the Antidepressant Activity of Selective Serotonin Reuptake Inhibitors Mediated by Nicotinic Acetylcholine Receptors?

**DOI:** 10.3390/molecules26082149

**Published:** 2021-04-08

**Authors:** Hugo R. Arias, Katarzyna M. Targowska-Duda, Jesús García-Colunga, Marcelo O. Ortells

**Affiliations:** 1Department of Pharmacology and Physiology, Oklahoma State University College of Osteopathic Medicine, Tahlequah, OK 74464, USA; 2Department of Biopharmacy, Medical University of Lublin, 20-093 Lublin, Poland; katarzyna.duda@umlub.pl; 3Departamento de Neurobiología Celular y Molecular, Instituto de Neurobiología, Campus Juriquilla, Universidad Nacional Autónoma de México, Querétaro 76230, Mexico; garciacolunga@unam.mx; 4Facultad de Medicina, Universidad de Morón, CONICET, Morón 1708, Argentina; mortells@retina.ar

**Keywords:** selective serotonin reuptake inhibitors, antidepressants, nicotinic acetylcholine receptors, noncompetitive antagonists, neuronal pathways, molecular modeling

## Abstract

It is generally assumed that selective serotonin reuptake inhibitors (SSRIs) induce antidepressant activity by inhibiting serotonin (5-HT) reuptake transporters, thus elevating synaptic 5-HT levels and, finally, ameliorates depression symptoms. New evidence indicates that SSRIs may also modulate other neurotransmitter systems by inhibiting neuronal nicotinic acetylcholine receptors (nAChRs), which are recognized as important in mood regulation. There is a clear and strong association between major depression and smoking, where depressed patients smoke twice as much as the normal population. However, SSRIs are not efficient for smoking cessation therapy. In patients with major depressive disorder, there is a lower availability of functional nAChRs, although their amount is not altered, which is possibly caused by higher endogenous ACh levels, which consequently induce nAChR desensitization. Other neurotransmitter systems have also emerged as possible targets for SSRIs. Studies on dorsal raphe nucleus serotoninergic neurons support the concept that SSRI-induced nAChR inhibition decreases the glutamatergic hyperstimulation observed in stress conditions, which compensates the excessive 5-HT overflow in these neurons and, consequently, ameliorates depression symptoms. At the molecular level, SSRIs inhibit different nAChR subtypes by noncompetitive mechanisms, including ion channel blockade and induction of receptor desensitization, whereas α9α10 nAChRs, which are peripherally expressed and not directly involved in depression, are inhibited by competitive mechanisms. According to the functional and structural results, SSRIs bind within the nAChR ion channel at high-affinity sites that are spread out between serine and valine rings. In conclusion, SSRI-induced inhibition of a variety of nAChRs expressed in different neurotransmitter systems widens the complexity by which these antidepressants may act clinically.

## 1. Introduction

Major depressive disorder (MDD), also known as clinical depression, is a neuropsychiatric disorder characterized by depressed mood, sleep and cognitive disturbances, anxiety, anhedonia or decreased interest in enjoyable activities, and suicidal thoughts, among other symptoms [1]. Although the etiology of MDD and related depression disorders is not fully understood, it is very complex and depends on neuronal, genetic, biological, cognitive, sociocultural, and psychological aspects [2,3,4].

Previous studies have determined that the malfunctioning of certain brain regions could be one important aspect in the development of MDD [5]. The most important brain abnormalities associated with depression are located in the prefrontal neocortex, limbic system, basal ganglia, and brain stem regions. In animal models of depression, as well as in depressed patients, the size and number of synapses are significantly reduced in several brain areas, including the nucleus accumbens (NAc), amygdala, cingulate cortex, hippocampus, and prefrontal cortex (PFC) [2,3,4].

Several hypotheses have been described and proposed in order to elucidate the etiology of MDD, but each one includes various neurotransmitter systems [6,7]. One of the first and most extensive depression hypotheses is associated with the decrease in extracellular neurotransmitter levels, such as serotonin (5-HT), dopamine (DA), and noradrenaline (NA) [8]. In this regard, substances that might elevate monoamine levels would be beneficial for depression treatment. Since the 1950s, MDD patients have been treated with monoamine oxidase inhibitors, which block monoamine reuptake and/or metabolism (e.g., selegiline) [9,10]. Among monoamine reuptake inhibitors, selective serotonin reuptake inhibitors (SSRIs; molecular structures are presented in Figure 1) have been classified as some of the most effective substances [7].

The cholinergic–adrenergic hypothesis proposes that the underlying cause of depression is an imbalance between the content of acetylcholine (ACh) and NA, producing an overstimulation of the cholinergic system over the noradrenergic system [11]. Cholinergic neurotransmission, which is mediated by muscarinic and nicotinic receptors, has been related to various neurophysiological processes, such as attention, learning and memory, mood, and appetite changes, as well as pathological conditions, including anxiety disorders and MDD [12,13,14]. Higher than normal extracellular levels of ACh and its hydrolysis metabolite, choline, have been observed in patients with depression [14,15,16]. Moreover, treatment with physostigmine, an acetylcholinesterase (AChE) inhibitor that increases the levels of synaptic ACh, elicited depression symptoms in humans [17,18,19] and depression-like effects in rodents [20].

Mounting evidence supports the view that other neurotransmitter circuits and pathways are involved in the clinical activity of SSRIs, including those modulated by nicotinic ACh receptors (nAChRs). This review focuses on functional, preclinical, therapeutic, and structural studies that support the activity of SSRIs on different nAChR subtypes.

## 2. Selective Serotonin Reuptake Inhibitors (SSRIs)

To date, the most accepted and best characterized mechanism of action for SSRIs is centered on the inhibition of serotonin transporters (SERT) located at presynaptic terminals. The first action of SSRIs is to inhibit SERT, increasing the synaptic levels of 5-HT up to seven-fold [21,22,23]. This elevated 5-HT concentration, in turn, activates a variety of postsynaptic 5-HT receptors in several brain areas, with result in changes in serotoninergic signaling, which is believed to attenuate depression conditions [24,25]. Furthermore, high extracellular concentrations of 5-HT trigger a negative feedback mechanism involving 5-HT_1A_ autoreceptors, which regulate 5-HT levels in the synaptic cleft. A high endogenous concentration of 5-HT may be sufficient for inhibiting nAChRs [26,27].

SSRIs are extensively used medications for the treatment of MDD and other neuropsychiatric maladies, including obsessive–compulsive disorders, panic disorders, alcoholism, obesity, migraines, and chronic pain [25,28,29]. For example, sertraline and paroxetine are prescribed for the treatment of panic attacks, obsessive–compulsive disorders, post-traumatic stress disorders, social anxiety disorders, and premenstrual dysphoric disorders, a severe form of premenstrual syndrome (https://www.webmd.com/corporate/default.htm). Fluoxetine is also employed in the treatment of eating disorders and slow-channel congenital myasthenic syndromes [29,30]. A common feature in both conditions is that of higher ACh concentrations compared to those in the normal population [20,30,31]. In slow-channel congenital myasthenic syndromes, high levels of ACh open slow-channel mutant nAChRs with long-lasting durations, allowing more cation influx into the endplate after an early onset of progressive muscle weakness [30,32]. Therefore, fluoxetine restores the pathological condition of congenital myasthenic syndromes by decreasing both the channel opening frequency and channel open time, as well as increasing the channel closed time.

Although SSRIs are used in many clinical conditions, the exact mechanisms underlying their clinical effects are not very clear. A plausible mechanism is that SSRIs, in addition to inhibiting SERT (e.g., fluoxetine inhibitory potency (IC_50_) = 6–17 nM [33,34,35]), modulate other targets, including a variety of nAChRs.

## 3. Nicotinic Acetylcholine Receptor Functions Modulate Depression States

nAChRs are members of the pentameric ligand-gated ion channel superfamily that also comprises 5-HT type 3, glycine, and γ-aminobutyric acid type A receptors. nAChRs are involved in a variety of physiological and pathological processes in both neuronal and non-neuronal tissues. In general, presynaptic nAChRs modulate the release of a variety of neurotransmitters, including those implicated in depression [36,37], whereas postsynaptic nAChRs mediate stimulatory transmission across the central and peripheral nervous systems [38,39].

The malfunctioning of nAChRs (by excessive or decreased activity) expressed in circuits regulating mood has been implicated in the development of anxiety states and MDD [15,40,41]. As part of the same concept, an imbalance between the cholinergic and noradrenergic systems—the so-called “cholinergic–adrenergic hypothesis of depression”—where the former system is more sensitive or more stimulated than the latter might cause MDD to develop [11]. In this regard, nAChR inhibition exerted by structurally different antidepressants, including SSRIs, could be relevant for their clinical efficacy [15].

The involvement of nAChRs in MDD has also been supported by studies using different brain imaging techniques. For instance, single-photon-emission computed tomography studies showed that the densities of β2* nAChRs (* indicates that other different subunits are potentially present) in the brains of patients with MDD and of healthy subjects were similar [42]. These results indicate that the number of nAChRs is not an important factor in the development of MDD. Nevertheless, a lower availability of β2* nAChRs was determined in the cortex, thalamus, cerebellum, striatum, hippocampus, amygdala, and brainstem in patients with either depression [42] or bipolar depression [43] compared to healthy subjects. This lower receptor availability could be caused by higher levels of endogenous Ach, which may result in receptor desensitization [42], a closed state that is unresponsive to ACh or other agonists. Moreover, positron emission tomography studies showed that Parkinson’s patients with depression symptoms had decreased levels of α4β2* nAChRs within subcortical regions, such as the putamen (involved in reward) and midbrain (where hypoactivity of 5-HT, DA, and/or NA has been related to MDD), as well as in cortical regions, including the anterior cingulate cortex and occipital cortex [44]. These data support the notion that altered nAChR-induced neurotransmission plays a role in the development of MDD.

## 4. Relationship between Smoking and Depression

Preclinical and clinical studies support the involvement of nAChRs in MDD [15,20,45,46]. Although the causality between nicotine exposure and depression remains unclear, clinical evidence supports a bidirectional association and dose-dependent relationship between smoking and depression [47]. For example, depressed patients smoke twice as much as the average population, probably to control their mood symptoms. People smoking more than twenty cigarettes per day double their risk of subsequent depression during their lives. Paradoxically, however, the chronic use of nicotine may lead to the development of depression symptoms, probably through a process involving nicotine-induced receptor desensitization [14,40,48,49]. This evidence highlights the correlation between smoking habits and increased risks of depression disorders, which, in turn, will increase the chances of using antidepressants in the long term.

A proposed mechanism by which smoking induces depression is that the excess of nicotine activates nAChRs and, consequently, the cholinergic signaling, thus breaking the balance between the cholinergic and noradrenergic systems [14,40,50,51]. Depression states, in turn, may increase the rate of smoking due to mood changes, which are positively correlated with the quantity of tobacco used. Additionally, chronic smoking leads to desensitization and upregulation of high-affinity β2* nAChRs (i.e., α4β2*, α6β2*) (see more details below), which, in turn, could contribute to depression symptoms [31,52]. In this regard, smokers and nonsmokers might respond differently to antidepressants, but the effects of smoking on treatment outcomes have rarely been examined and are therefore unclear. Depression symptoms in adults were more attenuated when an SSRI was combined with mecamylamine, a nonselective nAChR antagonist with antidepressant-like activity [53], compared to the SSRI alone [54]. Other analyses, however, have shown that smokers respond less to this drug combination than nonsmokers [55].

Additional studies also highlighted the importance of the cholinergic system in the development of depression. For example, augmenting ACh content by decreasing the AChE activity using the inhibitor physostigmine or virally delivered shRNAs produced depression-like behavior in rodents [20]. This is in accordance with the hypothesis that subjects with depression have higher endogenous ACh content, which may overstimulate the cholinergic system [42]. On the other hand, structurally different nAChR antagonists also produce antidepressant-like effects, probably by limiting nAChR function [14,45,46].

Anhedonia, a main symptom in major depression, has been linked to dysfunctions in the brain reward system, particularly when the DA content is decreased [56]. Therefore, a biochemical hypothesis of nicotine dependence [57] could also explain its relationship with nicotine-modulating neuronal pathways and depression. The initial and intrinsic biochemical basis of tobacco addiction should be found in the mechanisms by which nicotine promotes different effects on nAChRs. According to this model, nAChR desensitization and upregulation, but not activation, are compatible molecular processes that effectively cause nicotine dependence [58]. In addition to the classical desensitization process, low agonist concentrations can induce desensitization even without nAChR activation, a process called “high-affinity desensitization” [59]. This is a slow process that, in the prolonged presence of low concentrations of nicotine (~0.5 µM), preferentially affects α4β2* compared to α7* nAChRs. Nicotine accumulation during a smoking day is enough to produce nAChR desensitization. At nicotine concentrations present in chronic smokers’ brains, α4β2* nAChRs are preferably desensitized and subsequently upregulated in contrast to α7* nAChRs. Since presynaptic α4β2* nAChRs regulate GABA release onto dopaminergic neurons, a possibility is that nicotine-induced nAChR desensitization decreases GABAergic inhibition of dopaminergic neurons, and the consequent disinhibition finally promotes rewarding effects. After chronic nicotine use, however, the subsequent upregulation of α4β2* nAChRs enhances the inhibitory action of GABA on dopaminergic neurons, and consequently, even more nicotine is needed to restore at least a basal level of reward, leading to a vicious circle of nicotine-induced α4β2* desensitization (reward) and upregulation (enhanced reward inhibition).

To test this hypothesis, the effect of nicotine at smoking concentrations (0.3 µM) on GABAergic transmission was determined in pyramidal neurons from the medial PFC (mPFC), which modulates the brain’s reward circuitry [60]. The results clearly indicated that nicotine inhibits GABAergic postsynaptic currents by activating non-α7—probably α4β2* and α6β2*—nAChRs, whereas the role of α7* nAChRs in this process was less clear. These results contrasted with those of studies in the hippocampus, where α7* nAChRs were the main subtype involved in nicotine-induced modulation [61]. Our study also showed that nicotine increased membrane conductance, probably by activating postsynaptic α7* nAChRs, and decreased current frequency, corroborating the role of presynaptic β2* nAChRs in GABA release. In other words, our study supported the activation of pre- and post-synaptic α7* and β2* nAChRs in GABAergic modulation, while the possibility that nAChR desensitization is also involved in this process could not be ruled out [60].

There is also comorbidity between depression and anxiety conditions and nicotine addiction [41], highlighting the relationship between these psychiatric disorders and nAChR functioning. Although different antidepressants are successfully used in the treatment of both depression and anxiety, only bupropion and nortriptyline produced long-term beneficial effects when used for smoking cessation, whereas fluoxetine was effective only in smokers with strong depressive symptoms [62,63]. Moreover, non-selective antidepressants, such as monoamine oxidase inhibitors, did not produce benefits in smoking cessation therapy [62].

## 5. Preclinical Studies with Combinations of SSRIs and Nicotinic Ligands

There is experimental evidence showing that nicotinic ligands modulate the antidepressant-like activity of SSRIs. For instance, microdialysis analyses evidenced that nicotine elevates 5-HT levels in the frontal cortex during the initial 15 min, while nicotine in the presence of fluoxetine increases and prolongs the release of 5-HT for at least 2 h, and these effects were blocked by pretreatment with mecamylamine [64]. Additional studies determined that nicotine-evoked 5-HT release depends on α7 nAChR activation [46]. For instance, the combined administration of the α7-selective agonist PNU-282987 at a dose (30 mg/kg) that causes full receptor occupancy [65] with a subactive dose of citalopram (3 mg/kg), which inhibits ~50% of the SERT but is unable to induce antidepressant-like activity in mice per se [46,66], causes antidepressant-like effects. Since PNU-282987 alone showed no antidepressant-like activity, it is possible that the combination of α7 nAChR activation and partial SERT inhibition was responsible for the observed antidepressant-like effects, confirming nAChRs as targets for the treatment of major depression.

Although the administration of nicotine alone showed no effect in the tail suspension test (i.e., a very useful animal test to determine the antidepressant-like activity of drugs) and citalopram alone produced a slight decrease in immobility, the co-administration of both drugs resulted in a robust antidepressant-like activity [67]. Moreover, mecamylamine, but not dihydro-β-erythroidine (DHβE) (antagonist with relative selectivity for β2* nAChRs, including α4β2* and α6β2*), increased the antidepressant-like effect of citalopram [67], suggesting that these nAChR subtypes are not involved in the antidepressant activity of SSRIs. The latter coincides with Ca2+ influx experiments where citalopram had the lowest inhibitory potency for α4β2 nAChRs compared to other SSRIs [68] (Table 1). These results contrasted with the evidence that both mecamylamine and DHβE enhanced the antidepressant-like activity of imipramine (a tricyclic antidepressant that blocks both NE and DA reuptake transporters), suggesting that both functionally different antidepressants target distinct nAChR subtypes.

## 6. SSRIs Inhibit nAChRs at Clinical Concentrations

SSRIs inhibit a variety of nAChR subtypes with distinct potencies (Table 1) and by different mechanisms [26,27,30,70,71,74,77]. This functional inhibition may reduce the exacerbated cholinergic activity observed during depression conditions, which is consistent with the cholinergic hypothesis of depression [11,17,18,19,20].

Although Ca^2+^ influx results established that fluoxetine is the most potent SSRI in inhibiting a variety of nAChR subtypes, the use of different functional assays did not give only one value, but a range of values for each SSRI. Table 1 shows that the activity of each SSRI also depends on the studied nAChR subtype. For example, the potency of both fluoxetine and paroxetine was practically the same in the hα3β4 nAChR, while two-fold-lower values were determined for citalopram in the same receptor subtype. Considering a single nAChR subtype, the calculated IC_50_ values depended on the type of SSRI molecule. For example, the following inhibitory potency rank order was determined at hα4β2 nAChRs (IC_50_’s in µM): fluoxetine (4.4 ± 0.6) > paroxetine (8.6 ± 2.3) > citalopram (19.0 ± 4.2) (Table 1).

Although, in general, SSRIs inhibit nAChRs through noncompetitive mechanisms, the inhibition at the α9α10 nAChRs is mediated by a competitive mechanism. Voltage-clamp results revealed that citalopram inhibits ACh-evoked α9α10 currents in a voltage-independent and competitive manner [68]. This inhibitory mechanism was also observed in structurally different antidepressants, such as imipramine [79]. Since α9α10 nAChRs are not expressed in the brain, but preferably in peripheral immunocompetent cells [80], the inhibitory mechanism observed at this receptor is not relevant for the clinical efficacy of SSRIs in depression, but might be related to their anti-inflammatory activity [81].

An important aspect for determining whether SSRIs inhibit nAChRs at clinically relevant concentrations is the calculation of the plasma/brain ratio after drug treatment. The determined levels of SSRIs in plasma were generally lower than those in the brain. For example, after administration of 40 mg/day fluoxetine for 30 days, the plasma concentration could reach 0.29–0.97 µM, and in some patients, up to 1.6 µM [82,83]. A wide range of brain-to-plasma ratios (2.6–20) has been calculated for fluoxetine [84,85], which is concurrent with the relatively higher brain concentration (13 µM) [86]. Post-mortem studies in individuals that had been taking citalopram showed brain concentrations of 4.7 ± 3.3 µM, which was approximately four-fold higher than that in plasma [87]. Accordingly, it is feasible that different nAChR subtypes can be inhibited—at least partially—by SSRIs (Table 1). By meticulously determining both the inhibitory activity of SSRIs, such as fluoxetine, sertraline, paroxetine, and citalopram, at α4β2 nAChRs and the free brain level of each SSRI, a functional inhibition of only ~2.5% was calculated, a percentage that was 10-fold lower than that estimated to be necessary to produce a noticeable antidepressant-like activity [88]. In comparison, the calculated free mecamylamine concentration in the brain supported ∼20% inhibition of the α4β2* nAChR function [88]. Although these results suggest that α4β2 nAChRs are not important for the antidepressant-like activity of SSRIs, there are other nAChR candidates that might be involved. For instance, habenular α3β4* and hippocampal α7* nAChRs (Table 1), as well as α6β2* nAChRs, are expressed in the brain reward system. Although no information on the effect of SSRIs at α6β2* nAChRs is currently available, there is experimental evidence indicating that modulation of α6* nAChRs by LF-3-88, a novel partial agonist that would pharmacologically act like a competitive antagonist, induces antidepressant-like effects in several mouse models [89].

## 7. Neuronal Pathways Involved in the Antidepressant Activity of SSRIs

Since activation of presynaptic nAChRs modulates the release of several neurotransmitters involved in mood regulation, the malfunctioning of these receptors expressed in different neuronal pathways might be related to mood disorders [31,33]. Considering that SSRIs inhibit different nAChRs (Table 1), the simplest contemplated mechanism by which SSRIs might regulate nicotinic pathways involves the inhibition of presynaptic nAChRs. According to the cholinergic hypothesis of depression, SSRIs might recover the unbalance of cholinergic signaling by direct inhibition of presynaptic nAChRs expressed in brain regions associated with depression [15,90]. Nevertheless, new evidence supports the notion that the mechanism of action of SSRIs is more complex than we previously thought, and involves many neurotransmitter systems.

The dorsal raphe nucleus (DRN) is a serotoninergic center involved in physiological functions related to cognition, mood, and emotion. Glutamatergic hyperstimulation (e.g., under stress conditions), increasing 5-HT content, and the subsequent deficit at limbic projections have been considered as potential mechanisms that underlie depression conditions [91]. Thus, SSRI-induced inhibition of nAChRs expressed in DRN might decrease 5-HT content, thus alleviating this condition.

Figure 2 shows a scheme of a DRN serotoninergic neuron and its modulation by other neurotransmitter systems. Under normal conditions, glutamatergic excitatory afferents from pyramidal neurons of the mPFC directly activate the DRN, thus increasing extracellular 5-HT, which is, in turn, regulated by somatodendritic 5-HT_1A_ autoreceptors. Under stress conditions, however, glutamatergic neurotransmission is hyperactivated, flooding the DRN with 5-HT (i.e., so-called “5-HT flooding”), but attenuating 5-HT outflow. In fact, higher-than-normal levels of 5-hydroxyindoleacetic acid (5-HIAA; major 5-HT metabolite) have been found in the cerebrospinal fluid (CSF) of subjects with depression, corresponding to the 5-HT flooding hypothesis. An excessive stimulation of 5-HT_2A_Rs in the mPFC increases glutamatergic input, whereas an excessive stimulation of postsynaptic 5-HT_1A_Rs in the hippocampus is associated with a deficit at corticolimbic projection sites, thus compromising neuroplastic processes. Since NE and histamine are also increased during stress, 5-HT neurons are inhibited by NE transmission from the locus coeruleus and by histaminergic transmission.

Experimental results indicated that ~75% of the DRN neurons that project to the NAc, an area involved in brain reward, are serotoninergic, and are, in turn, modulated by nAChRs [90]. Activation (by nicotine) of postsynaptic nAChRs (i.e., α4β2* and α7* nAChRs) increased the firing frequency of DRN serotoninergic neurons, whereas activation of presynaptic α4β2* nAChRs augmented glutamate (Glu) release from excitatory glutamatergic afferents to the DRN [95,96,97,98]. Based on this scheme, it is possible that SSRIs alleviate depression symptoms through two processes (see Figure 2): (1) by inhibiting presynaptic nAChRs at glutamatergic afferents, which decreases 5-HT flooding at DRN neurons, and (2) by inhibiting postsynaptic nAChRs, which attenuates DRN activity. Activation of presynaptic α4β2* nAChRs also increased 5-HT release in serotoninergic projections to the NAc, which might be related to the anxiolytic and addictive activity of nicotine [93].

Other neurotransmitter systems modulated by nAChRs are also involved in depression, which adds to the complexity of SSRI-modulating actions. For instance, MDD is associated with reductions in inhibitory GABAergic transmission in the occipital cortex, anterior cingulate, and dorsomedial/dorsolateral PFC, as well as with an excessive increase in Glu release or changes in glutamate receptors (GluR )properties in the hippocampus and PFC [7,93]. There are contrasting explanations of how fluoxetine may improve information processing in these impaired neuronal networks. Komlósi et al. [94] suggested that fluoxetine can suppress excitatory glutamatergic transmission without changing the output of GABAergic neurons, whereas Van Dyke et al. [92] indicated that chronic fluoxetine potentiates excitatory synapses by activating 5-HT_1B_Rs, and may induce deleterious effects, such as long-term depression and decreased long-term memory. On the other hand, Méndez et al. [99] concluded that acute fluoxetine decreases GABAergic transmission in the hippocampus through presynaptic mechanisms, independently of its effect on SERT. Interestingly, cholinergic signaling via nAChRs in pyramidal cells and GABAergic interneurons modulates inhibitory circuits in the hippocampus, which is crucial for information processing [100]. Different SSRIs inhibit hippocampal nAChRs (Table 1), and this activity disinhibits GABAergic neurons, finally improving the symptoms in MDD. In other words, SSRI-induced inhibition of nAChRs expressed in both glutamatergic and GABAergic neurons—regulating excitatory and inhibitory responses, respectively—is considered fundamental in maintaining neuronal circuit oscillations, which seem to be disrupted during depression [101].

The rostromedial tegmental (RMTg) nucleus, which contributes to monoaminergic responses to stressing events and is involved in depression, receives cholinergic inputs from the laterodorsal and pedunculopontine tegmental nuclei that activate presynaptic α7 nAChRs, thus finally increasing Glu release in the lateral habenula (LHb). The LHb has been considered to play a role in the development of pain and depression, two highly prevalent disorders that coexist in ~60% of patients [102]. In turn, RMTg neurons exert an inhibitory effect mediated by GABA on dopaminergic neurons in the ventral tegmental area (VTA), which is part of the brain reward system [103,104]. In the VTA, nicotine increased postsynaptic current amplitude, which was blocked by methyllycaconitine, indicating the involvement of α7 nAChRs in the reward mechanism [103]. Based on this scenario, it is possible to suggest that SSRIs, by inhibiting α7 nAChRs, may decrease the inhibitory activity of GABAergic neurons in the reward system.

The LHb is connected to the limbic system, whereas the medial habenula (MHb) is connected to the interpeduncular nucleus. The habenulo–interpeduncular pathway is considered a second—not the least important—reward circuitry that modulates, directly and indirectly, the mesolimbic dopaminergic reward system. Interestingly, this cholinergic pathway is a major player in the aversive effects of nicotine [103,104]. In particular, the MHb expresses α3β4* nAChRs, which are considered targets for the anti-addictive activity of coronaridine congeners, including natural alkaloids, such as ibogaine and its active metabolite noribogaine, and the synthetic derivative 18-methoxycoronaridine. Our functional results showed that both coronaridine alkaloids and synthetic derivatives noncompetitively inhibit α3β4 nAChRs expressed in heterologous cells and the MHb [105,106]. It has been demonstrated that ibogaine and noribogaine induce antidepressant-like activity in rodents [107], probably through inhibition of habenular α3β4* nAChRs. Since citalopram also inhibits habenular α3β4* nAChRs [68] (Table 1), it is obvious to conclude that the observed inhibition may be involved, at least partially, in the clinical efficacy of SSRIs.

## 8. SSRIs Bind Distinct nAChR Subtypes with Different Affinities

Although we know that SSRIs inhibit different nAChR subtypes, the relationship between inhibitory potency, binding affinities, and clinical efficacy is less clear. Thus, the binding affinities of SSRIs and other antidepressants at various nAChRs in different conformational states were determined through radioligand competition experiments [70,77,108,109]. More specifically, the influence of several SSRIs on the maximal binding of either [^3^H]imipramine (a known tricyclic antidepressant and noncompetitive antagonist (NCA) of nAChRs [71,110]) to a variety of nAChR subtypes or [^3^H]TCP (a known NCA of *Torpedo* nAChRs [111]) to *Torpedo* nAChRs in the resting (toxin-bound) and desensitized (agonist-bound) states was determined. The studied SSRIs completely inhibited the specific binding of both [^3^H]imipramine [77] and [^3^H]TCP [70], respectively. The calculated binding affinities (K_i_) for desensitized nAChRs were generally higher than those in the resting state (Table 2). In the desensitized state, fluoxetine inhibited [^3^H]imipramine binding to α4β2 AChRs with higher affinity than with α3β4 (~5-fold) and α7 (~10-fold) nAChRs, respectively (Table 2). Comparing the binding affinity of different SSRIs, the following rank order was determined (K_i_’s in µM) for α4β2: fluoxetine (1.0 ± 0.1) > citalopram (4.1 ± 0.3) > paroxetine (6.7 ± 0.9), and for α3β4: citalopram (1.8 ± 0.1) > fluoxetine (4.8 ± 0.5) > paroxetine (6.9 ± 0.6), respectively (Table 2). These results support a direct interaction of SSRIs with NCA sites located in the nAChR ion channel (see Section 10).

## 9. SSRIs Interact with nAChRs in Different Conformational States

Although SSRIs generally inhibit nAChRs in a noncompetitive manner, the intrinsic inhibitory mechanisms are less clear. nAChRs exist in three allosteric transitions between different conformational states: a resting state, an active open-channel state, and several desensitized states, each with a non-conducting ion channel [38,112]. Even though the functional properties of SSRIs have not been studied on all existent nAChR subtypes and conformational states, the experimental evidence indicates that their activity is dependent on the conformational state. For instance, fluoxetine allosterically modulates nAChRs by interacting with the resting (closed), active (open), and desensitized (non-conducting) receptor states [27,30,70]. Particularly, fluoxetine promotes a faster nAChR desensitization and has stronger inhibitory potency at higher concentrations, suggesting a higher affinity to desensitized nAChRs [26,77] in accordance with the radioligand binding results (Table 2). This pharmacological characteristic is common to other structurally different antidepressants [110], suggesting a shared mode of action.

While the pharmacological activity of fluoxetine with nAChRs is subtype- and state-dependent, we can summarize its basic functional effects as follow: (a) Fluoxetine reduces the frequency of channel opening, (b) increases channel closed times, and (c) decreases channel open times, without (d) altering single-channel conductance [27,30]. Accordingly, fluoxetine activity at the whole-cell level results in increased nAChR desensitization and the reduction of agonist-induced current amplitude, ultimately reducing Ca^2+^ entry [27,30,69].

Based on this evidence, we can suggest a probable mechanism for the clinical activity of SSRIs. In depression conditions, where higher concentrations of ACh are expected, most nAChRs are likely in the desensitized state. As shown previously, this conformational (closed) state is powerfully inhibited by antidepressants and maintained for a longer time, thus synergistically reducing nAChR-related signaling, which may ultimately be beneficial for the improvement of MDD and other depression disorders.

## 10. Characterization of SSRI Binding Sites for Different nAChR Subtypes

Molecular docking simulations have been used to determine the molecular interactions of SSRIs with a variety of nAChR subtypes [70,77,108,109,113,114]. In this regard, based on the crystal structure of hα4β2 (PDB id: 5KXI) [115] and *Torpedo* nAChR molecular models (PDB id: 2BG9) [116], several neuronal nAChRs were constructed, including the α4β2, α3β4, α7, and α9α10 subtypes [68,70,108,109,113,117,118]. SSRIs and other antidepressants were subsequently docked to each nAChR model [68,70,117,118], and molecular dynamics simulations were performed to determine the stability of each interaction.

Docking simulations indicated that fluoxetine interacts within the middle portion of the ion channel of each nAChR subtype, except the α9α10 subtype. Considering that each subunit has one transmembrane M2 segment, the ion channel is formed by five M2 segments [119]. Although the ion channel is highly conserved among species, differences are also apparent among nAChR subunit sequences [119], producing variations in the nAChR ion channel structure. For example, in the α4β2 nAChR (Figure 3), the amino acid rings are named: outer or extracellular (position 20′), nonpolar (position 17′), valine (position 13′), leucine (position 9′), serine (position 6′), threonine (position 2′), intermediate (position -2′), and cytoplasmic or inner (position -5′), whereas in the α7 nAChR, the polar rings (positions 2′ and 6′) are switched: The threonine ring occurs at position 6′, and the serine ring at position 2′.

The molecular docking of fluoxetine suggested that this molecule interacts with residues lining the channel lumen located between amino acid rings from position 6′ up to position 13′ of each studied nAChR subtype [70]. The orientations of docked fluoxetine in the α4β2 (Figure 3) and α7 [76] nAChR models are essentially the same. The amino group and the aromatic ring containing the trifluoromethyl moiety of fluoxetine interact through van der Waals contacts with valine residues at position 13′. Moreover, the oxygen atom forms a hydrogen bond with the hydroxyl group provided by the α4-Ser251 residue at position 10′ (Figure 3).

The docking orientation of fluoxetine in the α3β4 nAChR ion channel is slightly different because the α4 subunit carries additional phenylalanine residues at position 13′, forming the valine/phenylalanine ring together with an α3 subunit. In particular, the position of the fluoxetine aminoaryl moiety in this ion channel is exchanged with the benzyl ring position that was suggested in other nAChR subtypes. Thus, both aromatic rings of the molecule bind to the three phenylalanine residues at position 13′ (i.e., β4-Phe253), probably through π–π interactions [117].

In the case of escitalopram (i.e., (S)-citalopram), two luminal binding sites at α3β4 nAChRs were characterized [68]. A high-affinity binding site was located between positions 5′ and 16′—in the middle of the ion channel—toward the extracellular ion channel’s mouth (Figure 4A). In addition, a low-affinity binding site for escitalopram was located closer to the cytoplasmic side, between positions –3′ and 6′. Although the high-affinity site corresponds to that found for fluoxetine in the α3β4 nAChR (see above), few differences between both antidepressants were detected. For instance, escitalopram, but not fluoxetine, established a cation–π interaction with α3-F255 (position 14′) and interacted with β4-T254 (position 12′) (Figure 4B). Interestingly, fluoxetine formed a π–π interaction with β4-Phe255 (position 13′), which coincides with that for coronaridine congeners [105]. The binding site for coronaridine congeners, located between positions 6′ and 13′ [105], overlaps with the site for SSRIs.

Since fluoxetine displaced [^3^H]imipramine binding to different nAChRs (Table 2), the molecular docking of imipramine was compared to that for fluoxetine at both α4β2 and α3β4 nAChRs [70,117]. The results suggested that imipramine interacts with a site located in the middle portion of the ion channel. A more detailed analysis indicated that in the case of α4β2 nAChRs, imipramine interacts with residues that form the valine and leucine, but not the serine, rings via van der Waals contacts [118], whereas in the α3β4 nAChR, imipramine interacts with several residues between the valine/phenylalanine (e.g., its aromatic rings form π–π interactions with three β4-Phe253 residues at position 13′) and leucine rings [117]. These docking results agree with radioligand binding data, suggesting that the SSRI binding site in different nAChR subtypes overlaps the imipramine locus in both desensitized and resting states [70,77].

Additional molecular docking studies using the α9α10 nAChR model indicated that escitalopram interacts with orthosteric binding sites, but not with luminal sites [68] (Figure 5A).

So far, this is the only nAChR subtype where the SSRI-induced inhibition is mediated by a competitive mechanism. The orthosteric sites for escitalopram are located at the interface between α10(+) (principal component) and α9(−) or another α10(−) (complementary component), but not at the α9(+) and α10(−) interface [68]. To study these interactions in more details, in silico mutations were performed on key residues at the orthosteric binding site, transforming α9α10 into α3β4 and vice versa. The docking results indicated that the mutations did not abolish escitalopram binding to the α9α10 mutant or enable escitalopram binding to the α3β4 mutant. To further investigate these intriguing findings, the role of the β9-β10 loop was assessed. In this regard, the orthosteric binding sites from α3, α9, and α10 were superposed with escitalopram docked at α10(+) (Figure 5B). The superposition showed that the β9-β10 loop, which contains the Cys pair typical of α-subunits, is widely open in α10, allowing escitalopram to fit into the binding site [68]. This loop is closer to the binding pocket in α9, and even closer in α3, compared to α10, causing escitalopram to overlap with the main-chain atoms of these subunits when it is docked to α10(+). Additional structural details may support the observed differences. More specifically, α10-R186 might force the β9-β10 loop in α10 to be set apart from the receptor agonist pocket (Figure 5C). Looking from the β9 sheet (Figure 5D), the positively charged α10-R186—in its extended mode in the model—points to the middle of the β9-β10 loop, contrary to the homologous hydrophobic α9-V186 and α3-Y184 residues. The homologous position at α7 nAChRs is another positively charged residue, α7-K182, but unlike to α10-R186, it is readily deprotonated when partially buried within proteins [120,121]. These structural differences might account for a unique orthosteric binding pocket at α9α10 nAChRs, which allows the accommodation of molecules larger than nicotine or Ach, as observed in α10 (Figure 5C). At the ion channel level, however, there is no structural difference that might explain the absence of SSRI-induced α9α10 nAChR ion channel blockade. It might be the case that an initial competitive inhibition of α9α10 nAChR prevails over channel blocking, but the latter mechanism can be reached at higher concentrations, as suggested in the experimental studies [68].

## 11. Concluding Remarks

Mounting evidence supports the idea that the therapeutic activity of many commonly used antidepressants, including SSRIs, is partially mediated through the inhibition of nAChRs, thus counteracting the overactivity of the cholinergic system, which leads to depressive symptoms. New data highlight the importance of nAChRs expressed in different neuronal pathways in the regulation of 5-HT-induced effects. DRN serotoninergic neurons have been studied with enough detail to ascribe a potential modulatory activity of nAChRs, especially under stress/depression conditions. The evidence that SSRIs also inhibit a variety of nAChRs expressed in different neurotransmitter systems increases the complexity by which these antidepressants may act clinically. Although “nicotinic modulation” of depression has been indirectly recognized, the development of specific ligands, such as inverse agonists and partial agonists, of therapeutic value is still lacking.

## Figures and Tables

**Figure 1 molecules-26-02149-f001:**
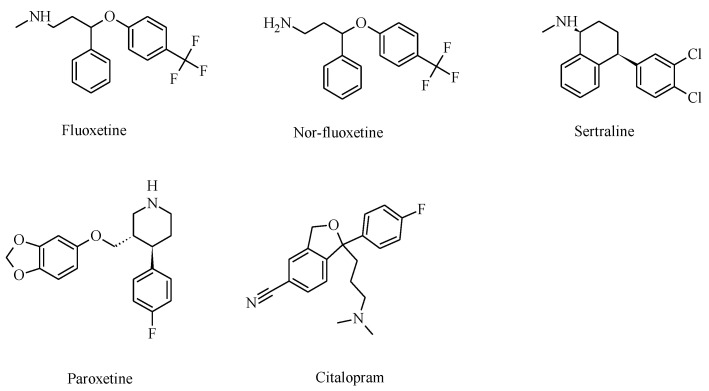
Molecular structures of clinically used selective serotonin reuptake inhibitors (SSRIs) that also inhibit various nicotinic acetylcholine receptor (nAChR) subtypes.

**Figure 2 molecules-26-02149-f002:**
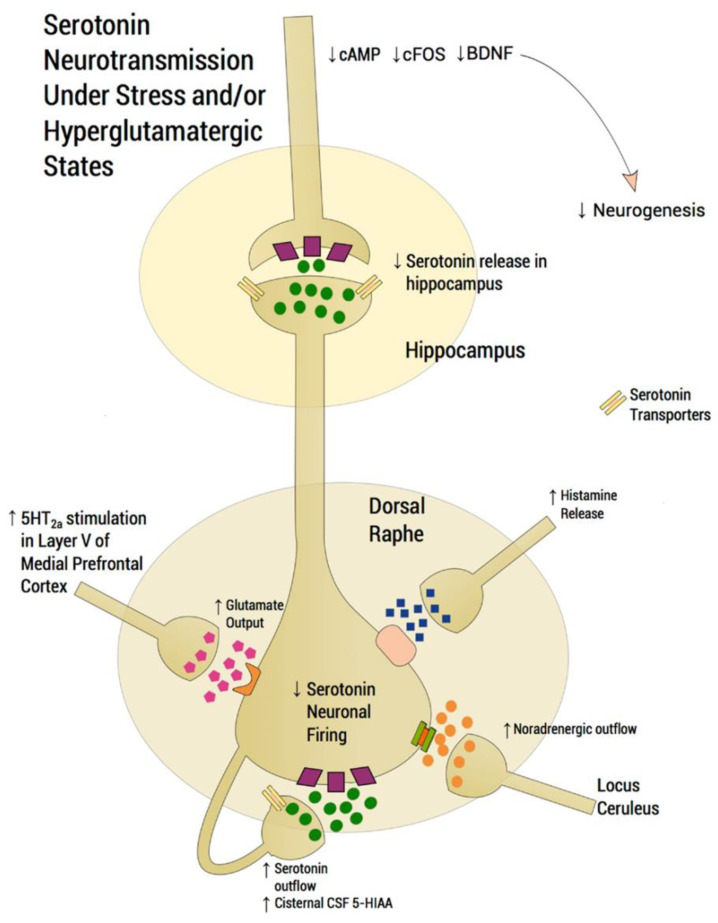
Scheme depicting the functioning of a dorsal raphe nucleus (DRN) serotoninergic neuron under stress conditions that induce depression disorders (modified from [91]). In normal conditions, glutamatergic excitatory afferents from pyramidal neurons of the medial prefrontal cortex (mPFC) directly activate the DRN, thus increasing extracellular serotonin (5-HT), which is regulated by somatodendritic 5-HT_1A_ autoreceptors, decreasing the excessive release in projection areas such as the mPFC and hippocampus. Under stress conditions, however, glutamatergic neurotransmission is hyperactivated, flooding the DRN with 5-HT (i.e., so-called “5-HT flooding”), but concomitantly attenuating 5-HT outflow. An excessive stimulation of 5-HT_2A_Rs in the mPFC increases glutamatergic input, whereas an excessive stimulation of postsynaptic 5-HT_1A_Rs in the hippocampus is associated with a deficit at corticolimbic projection sites, thus compromising neuroplastic processes. Based on this scheme, it is possible that SSRIs alleviate depression symptoms through two processes involving nAChRs: (1) inhibiting presynaptic nAChRs at glutamatergic afferents, which decreases 5-HT flooding to the DRN, and (2) inhibiting postsynaptic nAChRs, which attenuates DRN activity [7,92,93,94]. Since noradrenaline (NE) and histamine are also increased during stress/depression conditions, 5-HT neuron firing is inhibited by NE transmission from the locus coeruleus through α2-heteroceptors and by histamine transmission through histamine-1 receptors. Serotonin transporters are located presynaptically. CSF: cerebrospinal fluid; 5-HIAA: 5-hydroxyindoleacetic acid.

**Figure 3 molecules-26-02149-f003:**
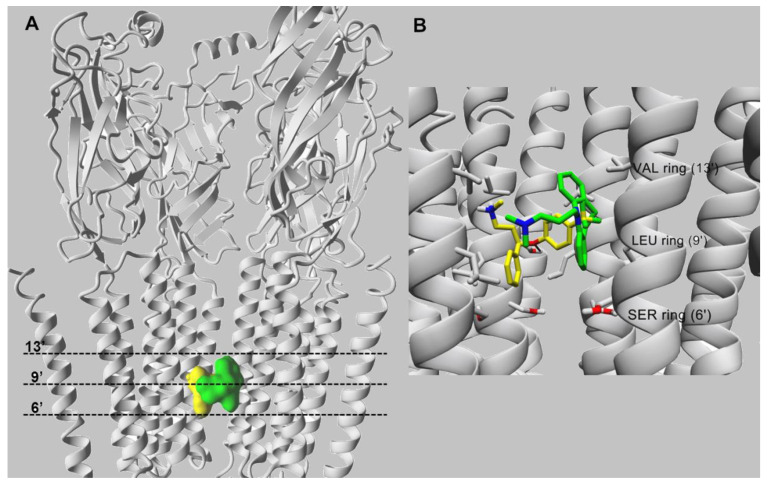
Molecular docking of fluoxetine (in yellow) and imipramine (in green), both in the protonated state, within the α4β2 nAChR ion channel (modified from [70]). (**A**) Side view of the overlapping binding sites for both ligands that interact with the middle portion of ion channel. (**B**) Imipramine (in green) and fluoxetine (in yellow) interact with the M2 transmembrane segments forming the lumen of the α4β2 nAChR ion channel. Both ligands interact mainly through van der Waals contacts with a domain formed between the valine (VAL) (position 13′) and serine (SER) (position 6′) rings. In addition, the black arrow indicates the hydrogen bond formed between the oxygen atom of fluoxetine and the hydroxyl group of α4-Ser251 (position 10′). For clarity, one β2 subunit is hidden. Residues involved in ligand binding are presented in stick mode (gray), whereas ligands are rendered either in the ball (**A**) or stick mode (**B**). All non-polar hydrogen atoms are hidden.

**Figure 4 molecules-26-02149-f004:**
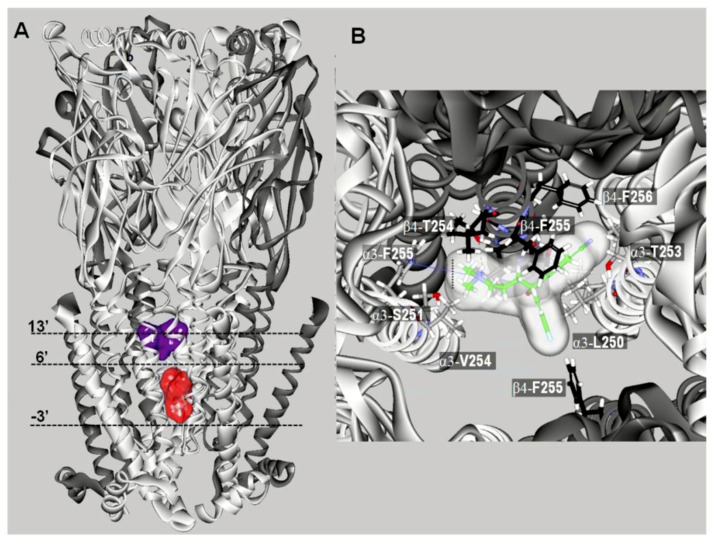
Docking sites for S-(+)-citalopram (escitalopram) in the (α3)_3_(β4)_2_ nAChR model (modified from [68]). (**A**) Escitalopram docked at two luminal sites (surface model): a high-affinity site located closer to the extracellular ion channel’s mouth (blue) and a low-affinity site located closer to the cytoplasmic side (red). The α3 (white) and β4 (dark gray) subunits are represented as solid ribbons. Dotted lines indicate the positions of the Gly (position -3′), Ser (position 6′), and Val (position 13′) rings along the ion channel. (**B**) Detailed view of escitalopram at the high-affinity luminal site, showing the cation–π interaction with α3-F255 (position 14′) and the interaction with β4-T254 (position 12′).

**Figure 5 molecules-26-02149-f005:**
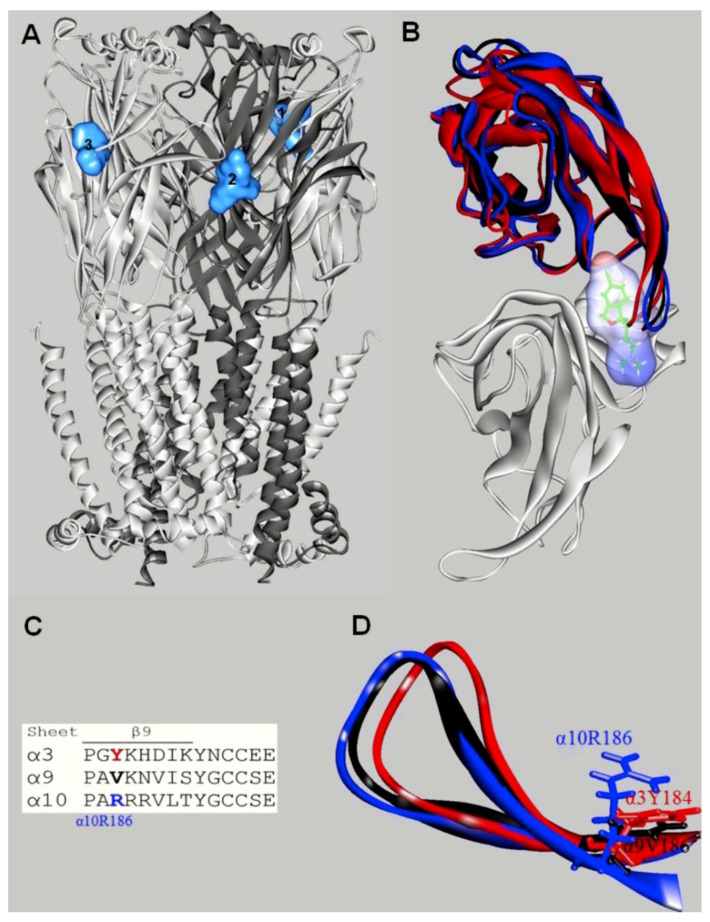
Docking sites for S-(+)-citalopram (escitalopram) in the h(α9)2(α10)3 nAChR model (modified from [68]). (**A**) Escitalopram (light blue surface model) interacted with three possible orthosteric sites located at the interface between the α10(+) (principal component) and α9(−) (or another α10(−)) (complementary component) subunits. The α10 (white) and α9 (dark gray) subunits are represented as solid ribbons. (**B**) Orthosteric binding sites at the superposed α3(+) (red), α9(+) (black), α10(+) (blue), and α9(−) (white) subunits. Escitalopram is shown as sticks surrounded by the molecular surface. The β9-β10 loop at the α3 and α9 subunits is closer to the receptor center than that at α10, and consequently, there is no room for escitalopram to fit in the agonist binding site in α3 and α9. The α3- and α9-β9-β10 loops overlap the ligand when it is docked as in the α9α10 receptor. (**C**) Amino acid sequence comparison between α3, α9, and α10 subunits at the level of the β9-β10 loop. Blue: Amino acids identified as causing different β9-β10 loop conformations. (**D**) Side-chain view showing differences in the occupied volume of side chains at the α10-R186, α9-V186, and α3-Y184 positions, respectively. The side-chain volume differences in the β9-sheet would force the α10-β9-β10 loop to be set apart from the binding pocket.

**Table 1 molecules-26-02149-t001:** Inhibitory potency (IC50) of SSRIs for different nAChR subtypes.

nAChR Subtype	SSRI	Method	IC_50_ (μM)	References
hα4β2	Fluoxetine	Voltage-clamp	1.83	[69]
		Ca^2+^ influx	4.4 ± 0.6	[70]
	Paroxetine	Ca^2+^ influx	8.6 ± 2.3	[70]
	Citalopram	Ca^2+^ influx	19.0 ± 4.2	[68]
rα2β4	Fluoxetine	Voltage-clamp	0.37 ^b^	[71]
			0.45 ± 0.03 ^c^	[27]
	Nor-fluoxetine	Voltage-clamp	0.13 ^b^	[71]
	Zimelidine	Voltage-clamp	0.67 ^b^	[71]
hα3β4	Fluoxetine	Ca^2+^ influx	2.0 ± 0.4	[70]
	Paroxetine	Ca^2+^ influx	2.6 ± 0.3	[70]
	Citalopram	Ca^2+^ influx	5.1 ± 1.3	[68]
rα3β4	Fluoxetine	Voltage-clamp	0.64 ± 0.03 ^c^	[27]
rα4β4	Fluoxetine	Voltage-clamp (+100–200 µM Zn^2+^)	0.3 ± 0.045 ^d^ 0.17 ± 0.018 ^d^	[72]
rα3β4α5	Fluoxetine	^86^Rb^+^ efflux	2.5	[73]
	Paroxetine	^86^Rb^+^ efflux	4.9	[73]
	Sertraline	^86^Rb^+^ efflux	3.1	[73]
hα7	Fluoxetine	Voltage-clamp	1.6	[74]
		Ca^2+^ influx	4.7 ± 0.83	[75]
		Ca^2+^ influx	4.9 ± 1.0	[70]
		Voltage-clamp	5.27	[69]
	Paroxetine	Ca^2+^ influx	8.6 ± 2.0	[70]
		Ca^2+^ influx	9.1 ± 0.6	[75]
	Sertraline	Ca^2+^ influx	10.0 ± 0.81	[75]
	Citalopram	Ca ^2+^ influx	18.8 ± 1.1	[68]
Hippocampal rα7 *	Nor-fluoxetine	Voltage-clamp	0.82 ± 0.04 ^a^	[76]
	Fluoxetine	Voltage-clamp	0.66 ± 0.06 ^a^	[76]
	Escitalopram	Voltage-clamp	28.9 ± 5.1	[76]
hα3β2	Fluoxetine	Voltage-clamp	4.14	[69]
rα9α10	Citalopram	Voltage-clamp	7.5 ± 0.9	[68]
mα1β1γδ	Fluoxetine	^86^Rb^+^ efflux	2.1	[73]
		Voltage-clamp (0 mV)	2.2	[26]
		Voltage-clamp (+100–200 µM Zn^2+^)	0.45± 0.04 ^d^ 0.19 ± 0.02 ^d^	[72]
		Voltage-clamp	0.22 ^b^	[71]
		Voltage-clamp	0.3 ^e^	[27]
hα1β1γδ	Fluoxetine	Ca^2+^ influx: 5 min pre-incubation 240 min pre-incubation 1440 min pre-incubation	1.8 ± 0.6 0.22 ± 0.01 0.17 ± 0.08	[77]
hα1β1εδ	Fluoxetine	Ca^2+^ influx	0.85 ^b^ 0.55 ^b^	[30]
mα1β1γδ	Nor-fluoxetine	Voltage-clamp	0.07 ^b^	[71]
hα1β1γδ	Paroxetine	Ca^2+^ influx: 5 min pre-incubation	4.8 ± 0.6	[77]
mα1β1γδ	Paroxetine	^86^Rb^+^ efflux	5.6	[73]
	Sertraline	^86^Rb^+^ efflux	3.5	[73]
Habenular mα3β4 *	Citalopram	Patch-clamp	7.6 ± 1.0	[68]
Hippocampal nAChRs	Citalopram	Nicotine-evoked NA release	0.93	[78]
	Fluoxetine	Nicotine-evoked NA release	0.57	[78]
nAChRs from superior cervical ganglion	Fluoxetine	Ca^2+^ influx	2.0	[69]

h, human. r, rat. m, mouse. * denotes the potential existence of other subunit(s). The ratio between the agonist-induced response in the presence of the antidepressant and the control agonist-induced response at the respective concentration of the antidepressant (^a^ 20, ^b^ 10, ^c^ 2, ^d^ 5, and ^e^ 1 µM, respectively).

**Table 2 molecules-26-02149-t002:** Binding affinities (K_i_) of SSRIs with resting and desensitized nAChRs.

Radioligand	nAChR Subtype	SSRI	K_i_ (μM) ^a^		Reference
			Resting State	Desensitized State	
[^3^H]Imipramine	hα4β2	Fluoxetine	3.2 ± 0.4	1.0 ± 0.1	[70]
		Paroxetine	16.1 ± 1.3	6.7 ± 0.9	[70]
		Citalopram	ND	4.1 ± 0.3	[68]
	hα3β4	Citalopram	ND	1.8 ± 0.1	[68]
		Fluoxetine	14.9 ± 1.4	4.8 ± 0.5	[70]
		Paroxetine	24.0 ± 1.8	6.9 ± 0.6	[70]
	hα7	Fluoxetine	16.9 ± 1.4	11.0 ± 1.0	[70]
		Paroxetine	8.9 ± 0.8	8.7 ± 0.6	[70]
[^3^H]TCP	*Torpedo* α1β1γδ	Fluoxetine	1.9 ± 0.2	0.96 ± 0.04	[77]
		Paroxetine	34 ± 2	2.5 ± 0.1	[77]

^a^ The inhibition constant (K_i_) represents the ligand affinity for the tested radioligand binding site(s) in the desensitized (agonist-bound) and resting (bungarotoxin-bound) states, respectively. h, human.

## Data Availability

The data presented in this study are available on request from the corresponding author.

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
