# Peer review of "Is the Antidepressant Activity of Selective Serotonin Reuptake Inhibitors Mediated by Nicotinic Acetylcholine Receptors?"

_molecules, 2021, doi:10.3390/molecules26082149_

Round 1

Reviewer 1 Report

REPORT ON MANUSCRIPT n. 1160807

The paper “Is the antidepressant activity of selective serotonin reuptake inhibitors mediated by nicotinic acetylcholine receptors?”, by H. R. Arias et al., is a review article dealing with the “anticholinergic contribution” of selective serotonin reuptake inhibitors (SSRIs) to their therapeutic utility as antidepressant drugs, a matter of debate since many years. Although the subject of the paper is not new, however the authors are competent in the field, the manuscript is well written and the text is supported by appropriate and updated references, with a detailed (I would say to some extent overdetailed) section focusing on the interaction at the molecular level of selected SSRIs with central and peripheral nicotinic acetylcholine receptor (nAChR) subtypes.

General comments

  1. The formula of Zimelidine should be deleted from Figure 1, since this drug was withdrawn from the marked in the mid-eighties, hence it cannot be cited among the “most recent and effective SSRIs”.
  2. Alignment of some lines in Table 1 should be improved.
  3. “Seronin Transporters” should be better localized in Figure 2. I wonder if their (a bit “neutral”) presence is necessary to support the concepts detailed in this figure’s caption.
  4. Given the complexity of the subject and several contradictory results on the role exerted by some nicotinic ligands, I would suggest that the authors underline in the “Concluding Remarks” that the “nicotinic modulation” of depression has been indirectly recognized but, at present, lacks specific ligands (such as inverse agonists/partial agonists) of therapeutic value.

Proposed language corrections

Page 1, line 30, ….. between residues 5’ and 13’ of the receptor protein sequence.

Page 2, line 52, …… increase monoamine levels …..

Page 2, line 63, …… and nicotinic receptors, ……

Page 3, lines 116-117, ….. could be relevant to their clinical efficacy.

Page 4, line 172, ….. are compatible molecular processes …..

Page 5, line 199 ….. add reference [60] at the end of the sentence.

Page 7, line 264, ….. were in general lower than those in the brain.

Page 7, line 270, ….. it is very feasible …..

Page 10, line 344, Other neurotransmitter systems ….. are also …..

Page 16, line 568, ….. also inhibit a variety of nAChRs …..

To summarize, I recommend publication of the manuscript in “Molecules” after a minor revision, in accordance with the above reported observations.

Author Response

Dear Reviewer,

Thank you for your comments. 

  1. The formula of Zimelidine should be deleted from Figure 1, since this drug was withdrawn from the marked in the mid-eighties, hence it cannot be cited among the “most recent and effective SSRIs”.

The formula of Zimelidine was deleted from the Figure 1.

  1. Alignment of some lines in Table 1 should be improved.

The Table 1 alignment was improved.

  1. “Seronin Transporters” should be better localized in Figure 2. I wonder if their (a bit “neutral”) presence is necessary to support the concepts detailed in this figure’s caption.

The “Seronin Transporters” location was indicated in Figure 2.

  1. Given the complexity of the subject and several contradictory results on the role exerted by some nicotinic ligands, I would suggest that the authors underline in the “Concluding Remarks” that the “nicotinic modulation” of depression has been indirectly recognized but, at present, lacks specific ligands (such as inverse agonists/partial agonists) of therapeutic value.

The following sentence was included in the Concluding Remarks: “Although “nicotinic modulation” of depression has been indirectly recognized, the development of specific ligands such as inverse agonists and partial agonists of therapeutic value is still lacking”.

Proposed language corrections

Page 1, line 30, ….. between residues 5’ and 13’ of the receptor protein sequence.

Page 2, line 52, …… increase monoamine levels …..

Page 2, line 63, …… and nicotinic receptors, ……

Page 3, lines 116-117, ….. could be relevant to their clinical efficacy.

Page 4, line 172, ….. are compatible molecular processes …..

Page 5, line 199 ….. add reference [60] at the end of the sentence.

Page 7, line 264, ….. were in general lower than those in the brain.

Page 7, line 270, ….. it is very feasible …..

Page 10, line 344, Other neurotransmitter systems ….. are also …..

Page 16, line 568, ….. also inhibit a variety of nAChRs ….

All suggestions were included in the text of the manuscript (in red).

Thank you for consideration of this submission.

Sincerely,

Katarzyna Targowska-Duda, Ph.D.

Department of Biopharmacy

Medical University of Lublin

4A Chodzki Street

20-093 Lublin, Poland

(+4881) 448-72-24

e-mail: katarzyna.duda@umlub.pl

Reviewer 2 Report

The review written by Arias et al. offers a comprehensive discussion of the role of nicotinic acetylcholine receptors (nAChRs) in mediating antidepressant activity of the selective serotonin reuptake inhibitors (SSRIs). This paper included a variety of studies, both clinical and experimental, substantiating interactions between SSRIs and nAChRs as a viable therapeutic target for the treatment of major depressive disorders. This paper is informative and provide a good summary of studies conducted to explore the potential associations between nAChRs and SSRIs. In my opinion, this paper is acceptable for publication following one minor revision.

Minor comment:
1. [Line 31] ...inhibition of a variety (of) nAChRs...

Author Response

Dear Reviewer,

Thank you for your comments.

The review written by Arias et al. offers a comprehensive discussion of the role of nicotinic acetylcholine receptors (nAChRs) in mediating antidepressant activity of the selective serotonin reuptake inhibitors (SSRIs). This paper included a variety of studies, both clinical and experimental, substantiating interactions between SSRIs and nAChRs as a viable therapeutic target for the treatment of major depressive disorders. This paper is informative and provide a good summary of studies conducted to explore the potential associations between nAChRs and SSRIs. In my opinion, this paper is acceptable for publication following one minor revision.

Minor comment:

  1. [Line 31] ...inhibition of a variety (of) nAChRs...

The suggestion was included in the text of the manuscript (in red).

Thank you for consideration of this submission.

Sincerely,

Katarzyna Targowska-Duda, Ph.D.

Department of Biopharmacy

Medical University of Lublin

4A Chodzki Street

20-093 Lublin, Poland

(+4881) 448-72-24

e-mail: katarzyna.duda@umlub.pl